# OpenReview forum: "Furina: Fragmented Uncertainty-Driven Refusal Instability Attack"
_ICML.cc/2026/Conference — ICML 2026 regular_

### Official Review · Reviewer_qe19 · 2026-03-13

**Soundness:** 3
**Presentation:** 3
**Significance:** 3
**Originality:** 3
**Overall Recommendation:** 4
**Confidence:** 4

**Summary:**

The submission assesses a general aspect of LLM safety alignment, namely the assumption that safety behavior operates as a near-binary threshold. The paper argues that safety refusals are governed by an intermediate instability band rather than a crisp decision boundary. To characterize this band, the authors propose a multi-metric diagnostic framework using external signals (token-level entropy H_tok, semantic entropy H_sem, ASR) and internal signals (HiddenDetect H_Dmax, Refusal Direction RD_max). They show that inputs in unstable regimes exhibit elevated output uncertainty paired with decreased internal safety activation, and argue this decoupling explains why detection-based defenses fail. Building on this, they introduce Furina, a jailbreak that induces this signature via fragmented, scene-anchored multi-turn prompts without model-specific optimization, evaluated on HarmBench and MM-SafetyBench across four commercial models.

**Compliance With Llm Reviewing Policy:**

Affirmed.

**Key Questions For Authors:**

I find the paper strong, and addressing weaknesses above (especially including error rates) would significantly add to its soundness.

**Limitations:**

Yes

**Strengths And Weaknesses:**

Strengths

* The multi-metric framework provides analytic value beyond the attack itself. Showing that diverse existing jailbreaks all produce the same decoupling signature despite their surface differences is a meaningful empirical finding. Table 2 is a useful cross-method comparison in this regard.

* The decoupling finding is concrete and somewhat surprising. RD_max dropping from 0.677 to 0.083 while ASR rises from 0% to 71% on LLaMA-2-7B directly challenges the assumption underlying detection-based defenses, and has clear practical implications.

* Extending the framework to MLLMs and showing the same diagnostic pattern holds across vision-language models is a useful contribution. The CLIP-similarity vs. safety-judgment plot in Figure 1 illustrates the phenomenon clearly.

* The ablation in Table 6 cleanly isolates the contribution of scene anchoring and semantic fragmentation, including a useful comparison against a progressive ladder baseline.

Weaknesses

* Results are reported as single point estimates throughout with no variance or confidence intervals. Given that the method is stochastic and some margins over baselines are modest, this makes it difficult to assess whether the improvements are reliable.

* The connection between the diagnostic framework and Furina's design is more narrative than mechanistic. The attack does not use the diagnostic metrics during prompt construction, there is no feedback loop steering generation toward higher H_tok or lower HD_max. The diagnostic and attack contributions are somewhat independent. Using the instability signature as an active optimization signal would be a natural direction for future work.

* The attack mechanics are adjacent to prior decomposition-based jailbreaks, and this relationship deserves more explicit acknowledgment. Fragmenting a harmful query into intent-preserving sub-questions across multiple turns has been explored before, and the paper would benefit from more carefully situating Furina relative to this line of work. The scene-anchoring component is a meaningful addition, but clearer positioning would help readers understand what is novel.

* The instability-band formalism introduces thresholds τ− and τ+ but never calibrates or operationalizes them. It is unclear how to determine in practice whether a given input falls in I versus S or U, or how sensitive the conclusions are to where these thresholds sit. As written, the partition functions more as a narrative device than a testable formalism.

* The central claim is that uncertainty amplification causes jailbreak success, but the evidence is correlational. Semantic rewrites shift both the semantic context and the entropy simultaneously, so it is not clear which is doing the work. An experiment that varies entropy independently of semantic content would be needed to support the causal reading.

* The diagnostic experiments are run on open-source models while the main attack results are on closed-source commercial APIs. The paper would benefit from showing the instability signature on the same models where the attack is evaluated, to more directly connect the two contributions.

* There is no discussion of what defenses the instability-band view suggests. If the signature is detectable at inference time, could it be used to build a more robust detector? Closing this loop would strengthen the paper's practical contribution.

* The GPT-4o judge is used without validation against human annotations. Given known disagreements between LLM judges on safety evaluations, some form of inter-rater check would improve confidence in the reported numbers.

The decoupling finding and the instability-band perspective are useful contributions to understanding safety failures. The practical implication for detection-based defenses is direct and the diagnostic framework could be a useful tool for future work.

---

> ### Author Rebuttal · Authors · 2026-03-31
>
> #### Reviewer qe19
>
> We thank the reviewer for the thorough and constructive feedback.
>
> **[W1] Variance and reliability of stochastic results.**
>
> We repeated the HarmBench evaluation 3 times on all four target models. Due to page limitation, the full results are in our response to Reviewer R9zZ's [W1] / [Q1].
>
> **[W2] Mechanistic link between the diagnostic framework and Furina.**
>
> Our framework clarifies what properties an effective attack context should satisfy, and Furina is designed to instantiate these properties. Concretely, entering the instability band requires three conditions: **semantic diffusion** to elevate token-level uncertainty, **contextual complexity** to weaken internal safety representations, and **cross-turn accumulation** to shift compliance probability.
>
> Furina's three-stage design maps directly onto these: fragmentation → (i), scene anchoring → (ii), multi-turn interaction → (iii).  We agree this link is principle-driven rather than feedback-driven; using instability metrics as active search signals is future work, and we will clarify this in the revision.
>
> **[W3] Positioning relative to prior decomposition-based jailbreaks.**
>
> We will position Furina more explicitly relative to prior multi-turn jailbreaks. **CoA** uses interrogation-style multi-turn attack chains[1], **Crescendo** gradually escalates from benign prompts by leveraging prior model responses[2], and **ActorAttack** use semantically related actors or clues to open multiple attack paths toward the same target[3].  Our claim is **not** that decomposition itself is new. Furina differs in using **fragmentation** to diffuse harmful intent toward the instability regime, together with **scene anchoring** to preserve cross-turn coherence without explicitly restating the full harmful workflow. We will clarify this positioning in the revision.
>
> [1] Chain of Attack: Hide Your Intention through Multi-Turn Interrogation, Findings of ACL 2025.
>
> [2] Great, Now Write an Article About That: The Crescendo Multi-Turn LLM Jailbreak Attack, USENIX Security 2025.
>
> [3] LLMs know their vulnerabilities: Uncover Safety Gaps through Natural Distribution Shifts, ACL 2025.
>
> **[W4] Conceptual vs. operational role of the instability-band formalism.**
>
> The $\tau^-$/$\tau^+$ partition is currently **conceptual rather than operational**. It is meant to formalize the transition from stable refusal to stable compliance through an intermediate stochastic regime, not to claim that we already have a calibrated classifier assigning inputs to $S$, $I$, or $U$. Because the boundary is high-dimensional and model-dependent while the inputs are discrete, explicit threshold calibration is nontrivial. In the current paper, the partition therefore serves as an **interpretive scaffold**, and we will clarify explicit threshold calibration as future work.
>
> **[W5] Causal interpretation of uncertainty amplification.**
>
> We agree that, in the semantic-rewrite setting, one should distinguish the effect of **semantic transformation** from that of **uncertainty amplification**. We provide partial evidence in **Table 2**: a suffix-based attack (**AmpleGCG**) appends an **optimized adversarial suffix** which is not constructed as an explicit semantic rewrite of the harmful request,  but still increases $H_{\text{tok}}$ (0.289→0.306) and $H_{\text{sem}}$ (0.091→0.138), while ASR rises from **0.08** to **0.24**. This suggests that **uncertainty amplification itself can already improve jailbreak success**, even without semantic rewriting.
>
> **[W6] Diagnostic models vs. attack evaluation models.**
>
> We partially bridge this gap with new **open-source** attack results: Furina reaches **89.0% ASR** on **Llama-2-7B-Chat** and **92.5% ASR** on **Qwen3-8B**, the same models used for diagnostics in Section 3. For **closed-source** APIs, only black-box signals such as $H_{\text{sem}}$ and ASR are observable, while internal metrics $H_{\text{tok}}$, HD_max​, RD_max are inaccessible. We will clarify this limitation and add external-metric diagnostics on commercial APIs in the revision.
>
> **[W7] Defensive implications of the instability-band view.**
>
> Our framework suggests two defense directions: **multi-turn semantic trajectory monitoring**, to detect when benign-looking fragments re-cohere into harmful intent, and **uncertainty-aware conservative policies**, where elevated generation-time uncertainty triggers stronger checking or constrained decoding. Both go beyond single-turn filtering. We will clarify this implication in the revision, while noting that a full defense design is beyond the current scope.
>
> **[W8] Human validation and judge reliability.**
>
> Three human annotators yielded ASR of 90.0%, 90.0%, and 88.0% on the same transcripts, closely matching GPT-4o (90.5%). We will add item-level agreement metrics in the revision. Full details in our response to Reviewer R9zZ's [W3] / [Q3].

---

> > ### Author Rebuttal · Reviewer_qe19 · 2026-04-03
> >
> > Thank you for the clarifications and additional experiments. I will keep my positive score.

---

> > > ### Author Response · Authors · 2026-04-04
> > >
> > > Thank you for your follow-up response and for your positive assessment of our rebuttal. We sincerely appreciate your time and consideration throughout the review process. We will continue to improve the paper in the revision stage. Thank you again, and we wish you all the best.

---

### Official Review · Reviewer_R9zZ · 2026-03-13

**Soundness:** 2
**Presentation:** 3
**Significance:** 3
**Originality:** 3
**Overall Recommendation:** 4
**Confidence:** 4

**Summary:**

This paper challenges the prevailing assumption that LLM safety operates as a near-binary threshold mechanism. The authors reveal that safety behavior is governed by an instability region where small perturbations induce stochastic refusal decisions rather than deterministic outcomes. They develop a multi-metric diagnostic framework combining external and internal signals (token-level entropy, semantic entropy, internal safety signals, ASR) to characterize this instability.

The key insight is that diverse jailbreak strategies share a common mechanism: amplifying uncertainty within instability zones. Building on this, the paper introduces Furina, a jailbreak attack that deliberately induces this signature through fragmented, scene-anchored prompts without model-specific optimization. The authors strive to address a core challenge in understanding LLM safety vulnerabilities. Overall, the authors consider a notable topic in AI safety and adversarial robustness.

**Compliance With Llm Reviewing Policy:**

Affirmed.

**Final Justification:**

I am maintaining my score of 4 (Weak Accept).

**Key Questions For Authors:**

1. Statistical Validation: Can you provide results with multiple runs and variance estimates? Given that GPT-4o judge has inherent randomness, how confident are you that the 8.5 percentage point improvement over ActorBreaker (90.5% vs 82.0%) is statistically significant?
2. Open-Source Model Validation: The diagnostic analysis uses LLaMA-2-7B and Qwen3-8B, but Furina is only tested on commercial APIs. Can you validate the attack effectiveness on open-source models to enable independent verification and reproducibility?
3. Human Validation of Judge: The paper acknowledges ASR values may vary under alternative scoring. Have you validated GPT-4o judgments against human evaluation? What is the inter-annotator agreement rate for the "unsafe" classification?

**Limitations:**

The paper includes an Impact Statement that appropriately discusses:
1. English-only benchmarks (HarmBench, MM-SafetyBench)
2. Limited to four commercial APIs, no validation on open-source models where results can be independently verified
3. Reliance on GPT-4o as judge, No human validation of GPT-4o judge reliability for these subtle semantic attacks
4. Single-run evaluation limits confidence in reported improvements

**Strengths And Weaknesses:**

Strengths:
1. Novel Conceptual Framework: The instability region concept challenges prevailing binary threshold assumptions about safety mechanisms. The unifying perspective that uncertainty amplification underlies diverse jailbreak strategies provides a valuable conceptual lens for understanding attacks.
2. Multi-Metric Diagnostic Framework: The paper develops a principled framework combining token-level entropy, semantic entropy, internal safety signals, and ASR to characterize instability. This provides practical tools for safety evaluation.
3. Cross-Modal Validation: The validation across both LLMs and MLLMs strengthens the generalizability claims. The semantic rewrite ladder experiment provides clear evidence of intermediate instability regions.

Weaknesses:
1. Single-Run Evaluation Without Variance Estimates: The paper runs each method once per query with no repeated runs, confidence intervals, or statistical significance tests. Given that GPT-4o judge itself has inherent randomness, the reported improvements (e.g., 90.5% vs 82.0% for ActorBreaker) could fall within noise margins. Without variance estimates, it's impossible to assess whether Furina's advantages are statistically meaningful.
2. Attack Only Validated on Commercial APIs: While the diagnostic analysis (Section 3) uses open-source models (LLaMA-2-7B, Qwen3-8B), the Furina attack itself is only tested on commercial black-box models (GPT-4o-mini, GPT-4o, Gemini, Claude). The claim of being a "transferable mechanism" lacks validation on open-source models where the community could independently verify and build upon results.
3. Judge-Dependent Results Without Human Validation: The paper acknowledges that "absolute ASR values may vary under alternative scoring schemes" but doesn't validate GPT-4o judgments against human evaluation. Given that the core contribution involves subtle semantic transformations, human agreement rates would strengthen confidence in reported ASR metrics.

---

> ### Author Rebuttal · Authors · 2026-03-31
>
> **[W1] / [Q1]** **Repeated runs and variance estimates.**
>
> We thank the reviewer for this important point. We agree that, given the stochastic components in the evaluation pipeline, reporting only single-point ASR estimates is insufficient to fully assess reliability.
>
> In response, we repeated the first 200 harmful queries of HarmBench evaluation **three times** for both Furina and ActorAttack on multiple commercial models. The repeated-run results are:
>
> | Target model | Method | Run 1 | Run 2 | Run 3 | Mean | Std |
> | --- | --- | --- | --- | --- | --- | --- |
> | **GPT-4o-mini** | **Furina** | 90.5 | 88.0 | 89.5 | **89.33** | **1.26** |
> |     | **ActorAttack** | 82.0 | 83.0 | 80.0 | **81.67** | **1.53** |
> | **GPT-4o** | **Furina** | 88.5 | 89.5 | 87.5 | **88.50** | **1.00** |
> |     | **ActorAttack** | 86.0 | 83.0 | 85.0 | **84.67** | **1.53** |
> | **Gemini-2.5-Flash** | **Furina** | 87.5 | 87.0 | 90.5 | **88.33** | **1.55** |
> |     | **ActorAttack** | 87.0 | 84.0 | 88.0 | **86.33** | **1.70** |
> | **Claude-4.5-Haiku-Thinking** | **Furina** | 72.0 | 69.0 | 71.0 | **70.67** | **1.25** |
> |     | **ActorAttack** | 67.5 | 65.0 | 63.0 | **65.17** | **1.84** |
>
> Across all four target models, Furina remains stronger on average than the compared multi-turn baseline. On **GPT-4o-mini**, the average gain is **7.66 points**, and notably the **lowest** Furina run (**88.0%**) still exceeds the **highest** ActorAttack run (**83.0%**), suggesting that the improvement is unlikely to be explained by judge noise alone. Similar stability is observed on the other commercial models, although we agree that three repetitions are still limited. We will include these repeated-run results in **mean ± std** form in the revision.
>
> **[W2] / [Q2] Open-source validation for reproducibility.**
>
> We thank the reviewer for this suggestion. We agree that open-source validation is important for reproducibility.
>
> In response, we added experiments on two open-source models using the first 200 harmful queries from **HarmBench**: **Llama-2-7B-Chat** and **Qwen3-8B**. Under the same evaluation pipeline with **GPT-4o as the judge**, Furina achieves **89.0% ASR** on **Llama-2-7B-Chat** and **92.5% ASR** on **Qwen3-8B**.
>
> These results show that Furina is effective not only on commercial APIs, but also on representative open-source models, supporting our claim that it captures a more general transferable mechanism. We note that these additional open-source experiments were added during the rebuttal period and are currently based on **single runs**. We will extend them with repeated-run variance estimates in the revision.
>
> **[W3] / [Q3] Human validation and judge reliability.**
>
> We thank the reviewer for this important point. We agree that judge-based ASR should be checked against human evaluation. In response, we added a human validation on the **same evaluation setting used for the main reported result** (rather than the repeated-run variance experiment), with **three independent human annotations**, which yield ASR estimates of **90.0%**, **90.0%**, and **88.0%**. These values are close to the GPT-judge result and support the reliability of the reported trend. We also tested several alternative LLM judges: **GPT-4o: 90.5%**, **GPT-3.5-Turbo: 89.5%**, **DeepSeek Judge: 95.0%**, and **Claude Haiku 4.5: 93.5%**. While the absolute ASR varies somewhat across judges, the overall conclusion remains unchanged, suggesting that the result is not an artifact of a single judge choice.
>
> At rebuttal time, we have validated agreement at the **aggregate ASR level**, but have **not yet computed item-level inter-annotator agreement** for the unsafe/safe labels. We agree this would further strengthen the evaluation and will add it in the revision.

---

> > ### Author Rebuttal · Reviewer_R9zZ · 2026-04-05
> >
> > Thank you for the thorough rebuttal; the repeated-run variance estimates, open-source model validation, and human judge agreement results have adequately addressed my concerns, and I am maintaining my score of 4 (Weak Accept).

---

> > > ### Author Response · Authors · 2026-04-06
> > >
> > > Thank you for your follow-up response and for your thoughtful consideration. We sincerely appreciate your time and support. We will continue to improve the paper in the revision stage. Thank you again, and we wish you all the best.

---

### Official Review · Reviewer_VwmG · 2026-03-24

**Soundness:** 3
**Presentation:** 3
**Significance:** 3
**Originality:** 3
**Overall Recommendation:** 4
**Confidence:** 4

**Summary:**

The paper proposes that safety behavior is governed by "instability regions" where small perturbations induce stochastic refusal decisions. They develop a multi-metric diagnostic framework combining external signals (token-level and semantic entropy) with internal signals (hidden-state safety activations) to characterize this instability, revealing a "decoupling phenomenon" where inputs in unstable regimes show elevated output uncertainty but decreased internal safety activation. Based on these insights, they introduce Furina, a black-box multi-turn jailbreak method that decomposes harmful queries into semantically drifted sub-questions, augmented with scene description and multimodal image context, to push the model into high-uncertainty regions.

**Compliance With Llm Reviewing Policy:**

Affirmed.

**Key Questions For Authors:**

1. What temperature and top-p settings were used? How many samples M were drawn per query?
2. How does Furina perform against existing defenses such as Llama Guard [1] or perplexity-based filters [2] or ETA[3]?
3. Can you provide causal evidence for the decoupling phenomenon? For example, does artificially suppressing refusal directions via activation patching yield comparable ASR gains without elevated entropy?

[1] Llama guard: Llm-based input-output safeguard for human-ai conversations, 2023.

[2] Detecting language model attacks with perplexity, 2023.

[3] ETA: Evaluating Then Aligning Safety of Vision Language Models at Inference Time. ICLR. 2025.

**Limitations:**

yes

**Strengths And Weaknesses:**

Strengths:
1. The paper offers a compelling perspective: diverse jailbreak strategies can be understood as mechanisms for amplifying uncertainty within instability zones.

2. The combination of external metrics (H_tok, H_sem, ASR) with internal safety signals (HD_max, RD_max) is well-motivated. The observed "decoupling phenomenon" where inputs in unstable regimes show elevated output uncertainty but decreased internal safety activation has significant scope for understanding why detection-based defenses may fail against sophisticated attacks.

3. The evaluation spans multiple models and spans both LLMs and MLLMs. And Table 6 provides comprehensive ablation that supports the importance of both fragmentation and scene anchoring.

Weakness

1. The "instability band" analysis lacks critical experimental details that undermine interpretability:Temperature, top-p, and nucleus sampling settings are never specified. Given that the entire framework rests on stochastic behavior, it is hard to assess whether observed "instability" is an intrinsic property of safety mechanisms or a result of high-temperature sampling. The ASR estimator (Equation 3) depends on M stochastic samples, but M is not reported.

2. The paper argues internal safety mechanisms weaken in unstable regions but does not test Furina against guard models, rejection sampling, or detection filters. This can help understand whether the insights translate to real-world defense evasion.

3. The analysis does not disentangle whether reduced RD_max drives jailbreak success or is a byproduct. Since RD_max is measured on outputs, it couples to model decisions, compliant outputs naturally exhibit lower refusal activation. The observed signature may redescribe ASR rather than explain its mechanism. Activation patching or similar interventions would strengthen causal claims

The paper is clearly written and  and offers a novel perspective on jailbreaks as uncertainty amplification with strong empirical results and comprehensive ablations support the core claims. However, missing sampling details and the causal ambiguity in the decoupling analysis weaken both reproducibility and interpretability.

---

> ### Author Rebuttal · Authors · 2026-03-31
>
> **[W1] / [Q1] Decoding details and robustness to sampling settings.**
>
> Thanks for the suggestion! All diagnostic experiments used nucleus sampling with $T{=}0.8$, top-p${=}0.9$, $M{=}8$ samples per query. We will add these to Section 5.1. To investigate whether the instability band is merely a high-temperature artifact. We reran the analysis on 100 AdvBench examples across $T \in \\{0.2, 0.5, 0.8\\}$ ($H_{\text{tok}}$ / $H_{\text{sem}}$ / ASR):
>
> | Rewrite | Llama3 (T=0.8) | Llama2 (T=0.8) | Qwen3 (T=0.8) | Llama3 (T=0.5) | Llama2 (T=0.5) | Qwen3 (T=0.5) | Llama3 (T=0.2) | Llama2 (T=0.2) | Qwen3 (T=0.2) |
> |---|---|---|---|---|---|---|---|---|---|
> | O | 0.665/0.367/0.09 | 0.342/0.086/0.00 | 0.235/0.096/0.08 | 0.509/0.271/0.05 | 0.328/0.067/0.00 | 0.226/0.067/0.08 | 0.441/0.103/0.05 | 0.290/0.039/0.00 | 0.231/0.042/0.06 |
> | M | 0.660/0.371/0.09 | 0.340/0.095/0.00 | 0.234/0.091/0.08 | 0.516/0.279/0.10 | 0.331/0.072/0.01 | 0.231/0.072/0.08 | 0.455/0.147/0.07 | 0.332/0.067/0.00 | 0.225/0.037/0.08 |
> | MD | 0.626/0.309/0.25 | 0.368/0.101/0.04 | 0.264/0.106/0.31 | 0.514/0.242/0.15 | 0.351/0.079/0.06 | 0.260/0.083/0.27 | 0.461/0.162/0.11 | 0.348/0.080/0.06 | 0.260/0.044/0.26 |
> | H | 0.711/0.305/0.66 | 0.405/0.144/0.26 | 0.315/0.132/0.51 | 0.565/0.236/0.50 | 0.389/0.116/0.24 | 0.313/0.103/0.49 | 0.546/0.133/0.42 | 0.368/0.067/0.25 | 0.315/0.060/0.43 |
> | S | 0.891/0.182/0.88 | 0.435/0.141/0.71 | 0.334/0.119/0.73 | 0.804/0.153/0.80 | 0.437/0.116/0.72 | 0.331/0.097/0.61 | 0.833/0.121/0.75 | 0.421/0.087/0.71 | 0.330/0.063/0.58 |
>
> The monotonic ASR progression O→S persists at every temperature. For example, Llama2 at $T{=}0.2$: ASR follows 0.00→0.00→0.06→0.25→0.71. Lower temperatures compress absolute magnitudes but do not remove the instability pattern, confirming it reflects a structural property of safety alignment rather than a sampling artifact.
>
> **[W2] / [Q2] Performance against existing defenses.**
>
> We tested Furina on 200 HarmBench queries against GPT-4o-mini with two defense families, reporting both **input-side intercept rate** and **end-to-end defended ASR**:
>
> - **Llama Guard**: intercept rate **0.0%**, defended ASR **77.0%**. Furina's sub-questions are individually benign, so a per-turn guard classifier finds nothing to block.
> - **Perplexity filter** (threshold at p95 = 197.15 on 1,680 benign queries): intercept rate **2.5%** end-to-end, defended ASR **89.0%**. Furina's fragments are natural-language questions, not adversarial strings, so they do not trigger perplexity anomalies.
>
> These results empirically validate our Section 3.3 prediction: defenses that rely on detecting "unsafe-looking" inputs fail precisely because, in the instability band, internal safety representations are already suppressed. We were unable to complete ETA evaluation due to dependency conflicts in the released code, and will include it in the revision.
>
> **[W3] / [Q3] Causal role of $\text{RD}_{\text{max}}$ and activation patching.**
>
> **Clarifying the measurement.** $\text{RD}_{\text{max}}$ (Eq. 7) is computed from **input-side intermediate-layer activations**, not from output tokens—it captures the model's internal prompt representation *before* generation.
>
> **Activation patching tests the causal question directly.** We suppressed the refusal direction in the last 4 layers of Llama-2-7B-Chat during inference:
>
> | Condition | RDmax clean | RDmax intervened | Δ RDmax | ASR normal | ASR intervened | Δ ASR |
> |---|---|---|---|---|---|---|
> | Original | 0.5642 ± 0.0376 | 0.1870 ± 0.0375 | −0.3772 | 0.14 | 0.16 | +0.02 |
> | High rewrite | 0.4432 ± 0.1218 | 0.1666 ± 0.0575 | −0.2765 | 0.38 | 0.34 | −0.04 |
>
> As shown above, the intervention successfully suppresses RDmax​ (by 0.28–0.38) but produces **negligible ASR change** (Δ ASR within ±0.04). This directly answers the question: artificially suppressing the refusal direction does *not* yield ASR gains. The causal arrow from low RDmax​ to high ASR does not hold.
>
> In fact, our paper does **not** claim RDmax​ reduction *causes* jailbreak success but to explain **why detection-based defenses fail**: methods like HiddenDetect[3] assume unsafe inputs produce distinguishable safety representations, but in the instability band, these representations weaken (RDmax​↓) precisely when the model is most vulnerable (ASR↑). The patching result confirms these are **two parallel effects** of entering the instability band—high-entropy contexts simultaneously disrupt safety representations and destabilize generation—rather than a causal chain from one to the other. $\text{RD}_\text{max}$​ is the mechanism of **defense evasion**, not of jailbreak success itself. We will revise Section 3.3 to state this more precisely.
>
> [1] Refusal in Language Models Is Mediated by a Single Direction, NeurIPS 2024.
>
> [2] How to use and interpret activation patching, arXiv 2024.
>
> [3] HiddenDetect: Detecting Jailbreak Attacks against Large Vision-Language Models via Monitoring Hidden States, ACL 2025.

---

> > ### Author Rebuttal · Reviewer_VwmG · 2026-04-03
> >
> > Thank you for the rebuttal. The additional experimental results address my concerns. I will maintain my current score.

---

> > > ### Author Response · Authors · 2026-04-04
> > >
> > > Thank you for your follow-up comment and for your encouraging assessment of our rebuttal. We greatly appreciate your time and thoughtful consideration during the review process. We will continue to revise and strengthen the paper in the next stage. Thank you again, and we wish you all the best.

---

### Decision · Program_Chairs · 2026-04-30

**Decision:**

Accept (regular)

**Comment:**

This paper studies multi-turn jailbreaking for LLMs and MLLMs. It first demonstrates that models exhibit instability near refusal boundaries, then develops a multi-metric diagnostic framework combining external and internal signals. Finally, it introduces Furina, a jailbreak attack that induces high-entropy inputs to exploit this instability.

Reviewers appreciate the novel mechanism for understanding AI safety, the multi-metric diagnostic framework to characterize instability, and the extensive evaluation across both LLMs and MLLMs.

During the rebuttal, the authors provided several new results, including performance against existing defenses, robustness across key hyperparameters, repeated runs with variance estimates, and human validation of judge reliability. This addressed most concerns.